# A Mouse Model That Mimics AIDS-Related Cytomegalovirus Retinitis: Insights into Pathogenesis

**DOI:** 10.3390/pathogens10070850

**Published:** 2021-07-06

**Authors:** Jay J. Oh, Jessica J. Carter, Richard D. Dix

**Affiliations:** 1Viral Immunology Center, Department of Biology, Georgia State University, Petit Science Building, 161 Jesse Hill, Atlanta, GA 30303, USA; joh17@gsu.edu (J.J.O.); jcarter80@gsu.edu (J.J.C.); 2Department of Ophthalmology, Emory University School of Medicine, Atlanta, GA 30303, USA

**Keywords:** human cytomegalovirus, murine cytomegalovirus, retinitis, AIDS, MAIDS, mouse model

## Abstract

With the appearance of the worldwide AIDS pandemic four decades ago came a number of debilitating opportunistic infections in patients immunosuppressed by the pathogenic human retrovirus HIV. Among these was a severe sight-threatening retinal disease caused by human cytomegalovirus (HCMV) that remains today a significant cause of vision loss and blindness in untreated AIDS patients without access or sufficient response to combination antiretroviral therapy. Early investigations of AIDS-related HCMV retinitis quickly characterized its hallmark clinical features and unique histopathologic presentation but did not begin to identify the precise virologic and immunologic events that allow the onset and development of this retinal disease during HIV-induced immunosuppression. Toward this end, several mouse models of experimental cytomegalovirus retinitis have been developed to provide new insights into the pathophysiology of HCMV retinitis during AIDS. Herein, we provide a summary and comparison of these mouse models of AIDS-related HCMV retinitis with particular emphasis on one mouse model developed in our laboratory in which mice with a murine acquired immunodeficiency syndrome (MAIDS) of murine retrovirus origin develops a reproducible and well characterized retinitis following intraocular infection with murine cytomegalovirus (MCMV). The MAIDS model of MCMV retinitis has advanced the discovery of many clinically relevant virologic and immunologic mechanisms of virus-induced retinal tissue destruction that are discussed and summarized in this review. These findings may extend to the pathogenesis of AIDS-related HCMV retinitis and other AIDS-related opportunistic virus infections.

## 1. Introduction

HIV is a pathogenic human retrovirus that causes unique immunosuppression characterized by a progressive deterioration of immune functions that culminates in the clinical condition known as AIDS [1,2]. A central feature of AIDS is the development of a wide spectrum of diseases that arise due to the appearance of a number of opportunistic pathogens that flourish within the host due to loss of various components of adaptive immunity, most notably cellular immunity. Among AIDS-related opportunistic pathogens are human herpesviruses that include human cytomegalovirus (HCMV), a ubiquitous b-herpesvirus that infects ~80% of adults worldwide as determined by seropositivity [3,4,5]. Primary HCMV infection that usually occurs during childhood is often asymptomatic with the virus eventually establishing a life-long latent infection within bone marrow and circulating monocytes [6,7]. During HIV-induced immunosuppression, however, the virus reactivates and may then cause systemic disorders such as hepatitis and colitis. HCMV may also exhibit neurotropic properties and invade the central nervous system and retina to cause life-threatening encephalopathies and other neurologic disorders [8,9] as well as a sight-threatening retinitis [10,11], respectively. 

AIDS-related HCMV retinitis initially caused vision loss and blindness in ~30% of AIDS patients either unilaterally or bilaterally prior to the therapeutic use of combination antiretroviral therapy (ART) [12,13,14]. The therapeutic use of ART to manage HIV infection directly and thereby prevent the appearance of opportunistic viruses has resulted in a significant decrease in the incidence of AIDS-related HCMV retinitis within the United States. This retinal disease affecting both males and females without sex differences remains an ophthalmologic problem in other parts of the world, however, due to a growing number of HIV-infected persons who do not have access to ART or who fail to respond to ART [15,16]. 

Four decades after the appearance of AIDS in our society, and the recognition of AIDS-related HCMV retinitis worldwide, a large knowledge base has been established that defines with clarity its clinical and histopathologic presentations [17,18,19]. The virologic, immunologic, and pathogenic events that lead to the onset and development of HCMV retinitis in the unique setting of retrovirus-induced immunosuppression of AIDS has nonetheless remained unclear. 

## 2. Mouse Models of Experimental Cytomegalovirus Retinitis

Cytomegaloviruses are highly species-specific [20,21]. Thus, new and necessary information on the pathogenesis of HCMV retinitis under various immunosuppressive conditions has been provided using a number of experimental mouse models of this retinal disease. Due to the extreme species-specificity of HCMV, development of these experimental mouse models has relied on murine cytomegalovirus (MCMV), a mouse b-herpesvirus that shares many similarities with HCMV. These include similarities in genomic structure, cellular and tissue tropisms, and immunologic responses by the host [22,23]. 

Early attempts to create an animal model for HCMV retinitis using immunologically normal mice of various strains were largely unsuccessful with detection of only MCMV antigens and/or MCMV DNA within various ocular tissues following anterior chamber or intravitreal virus inoculation but without development of retinitis [24,25,26]. Holland’s laboratory [27] ultimately succeeded in developing an experimental mouse model of focal retinal necrosis in Swiss Webster mice immunosuppressed by cyclophosphamide treatment following intraocular inoculation with MCMV via intravitreal injection. Although histopathologic features consistent with HCMV retinitis in immunocompromised humans were reported that included the appearance of cytomegalic cells and virus inclusions within retinal cells and foci of full-thickness retinal necrosis, only a small number of the mice developed this pattern of retinal histopathology, and a majority of the animals succumbed to systemic MCMV disease. 

A more reproducible experimental mouse model of HCMV retinitis in immunocompromised humans was later developed by Atherton, Cousins, and coworkers [28,29] that has been used extensively to investigate the pathogenesis of this retinal disease during conditions of immunosuppression. Unique to this animal model is introduction of MCMV into the eye of immunosuppressed BALB/c mice via careful injection of virus into the supraciliary space rather than injection of virus into the anterior chamber or vitreous cavity of the eye. This inoculation route results in the delivery of infectious virus into the peripheral choroid, retinal pigmented epithelium (RPE), and/or outer retinal layers of the neurosensory retina and culminates in a full-thickness necrotizing retinitis in 80 to 100% of animals by 10 days after virus infection in mice immunosuppressed by corticosteroid (methylprednisolone) treatment [30,31]. Importantly, none of these animals become ill or die due to systemic MCMV infection. Moreover, the retinal disease that develops exhibits histopathologic features similar to those observed during AIDS-related HCMV retinitis. These include areas of full-thickness retinal necrosis with transition zones that separate areas of involved retina from areas of uninvolved retina, foci of cytomegalic cells, and intracellular virus inclusions within RPE and retinal cells [32,33]. While arguably producing a clinically relevant retinal disease, this experimental mouse model of MCMV retinitis does not faithfully reproduce the unique pathophysiology of HIV-induced AIDS. Unlike HIV-induced AIDS, corticosteroid treatment induces a global immune suppression involving a systemic loss of innate and adaptive immune responses within a matter of days. Thus, the retinal disease that develops in corticosteroid immunosuppressed BALB/c mice following supraciliary MCMV inoculation is devoid of macrophages and neutrophils even though these cells of innate immunity contribute significantly to the inflammation associated with the development of HCMV retinitis in patients with AIDS [28,34]. Of greater significance, however, is the inability of corticosteroid immunosuppression to reproduce the unique HIV-induced shift in cytokine production from a Th1 profile to a Th2 profile that is a hallmark pathogenic feature of AIDS [35,36]. 

In an attempt to create a more clinically relevant mouse model of AIDS-related HCMV retinitis, we employed C57BL/6 mice with murine acquired immunodeficiency syndrome (MAIDS) that develops within 1 to 3 weeks following systemic infection with the murine retrovirus mixture, LP-BM5 [37,38,39]. The progression of MAIDS in mice reproduces many of the pathogenic features of AIDS in humans. These include the appearance of persistent generalized lymphadenopathy, polyclonal B cell activation and hypergammaglobulinemia, and ultimately dysfunction of CD4+ T cells and CD8+ T cells by 8 weeks after immunosuppressive retrovirus infection [38,40]. Most importantly, however, is the appearance of a prominent Th1/Th2 shift in cytokine production that commences at ~3 weeks after retrovirus infection [38,40], a feature of MAIDS that makes it a particularly attractive mouse model of AIDS in humans. Nonetheless, like all animal models of disease, MAIDS also exhibits some pathogenic features not consistent with AIDS. Whereas macrophages, T cells, and B cells of mice with MAIDS are all infected with the murine retrovirus mixture, B cells are excluded as targets for HIV infection during AIDS [38,40]. B cells being a target for murine retrovirus infection during MAIDS also results in the development of a prominent splenomegaly due to the accumulation of B cells and macrophages within the spleen of mice with MAIDS, a pathogenic feature not seen during AIDS [38,40]. 

Building on the early development and comprehensive characterization of MAIDS by Morse and colleagues [37,38,39], we discovered that the intraocular inoculation of mice with MAIDS of 8 to 10 weeks duration with MCMV by the subretinal route (a variation of the supraciliary route) results in the development of a reproducible retinitis at 10 days after inoculation and without animals dying due to systemic dissemination of virus. The retinal disease that develops is characterized by full-thickness retinal necrosis associated with the appearance of foci of cytomegalic cells and virus inclusions within RPE and retinal cells as seen during retinitis development within the eyes of MCMV-infected corticosteroid-immunosuppressed mice. In sharp contrast, however, the retinitis within eyes of MCMV-infected mice with MAIDS is also associated with a significant infiltration of inflammatory cells consisting of macrophages and neutrophils [29,41] as seen during AIDS-related HCMV retinitis [29,38,41]. Our laboratory has therefore used the MAIDS model of MCMV retinitis to continue to define the complex virologic, immunologic, and pathogenic events that work either separately or collectively to allow the onset and development of cytomegalovirus retinitis in the unique setting of retrovirus-induced immunosuppression.

## 3. Insights into Pathogenesis

Several insights into the pathogenesis of MAIDS-related MCMV retinitis have been discerned by us using this experimental mouse model of AIDS-related HCMV retinitis, some predicted, yet many others unexpected, as summarized below. The reader is also directed to other reviews of the MAIDS model of MCMV retinitis for more detailed information [5,14,38,42]. 

### 3.1. Virus Replication and Cytopathology

Like many animal viruses, cytomegaloviruses induce a number of host cell changes during productive replication that lead to cell injury or cell death and thereby contribute to tissue pathology and the appearance of clinical disease. That virus replication and cytopathology contribute to the onset and development of retinal tissue destruction during the pathogenesis of AIDS-related HCMV retinitis has been recognized for many years. Evidence for direct virus infection of the retinal tissues of the eyes of AIDS patients with HCMV retinitis include the detection of HCMV-encoded nucleic acids and proteins, the detection of virus-induced inclusions within various retinal cells, and the appearance of pathognomonic cytomegalic cells within areas of retinal necrosis [19,29]. These criteria for virus replication and cytopathology within retinal tissues as a mechanism for the pathogenesis of AIDS-related HCMV retinitis were documented during early characterization of the MAIDS model of MCMV retinitis [29]. Significantly high amounts of infectious virus (~30,000 infectious virus particles per eye) were also recovered from the MCMV-infected eyes of mice with MAIDS of 10 weeks duration (MAIDS-10 mice) who are susceptible to retinitis after subretinal MCMV inoculation. In sharp contrast, only measurable but considerably lower amounts of infectious MCMV (~250 infectious virus particles per eye) were recovered from the MCMV-infected eyes of healthy mice who are resistant to retinitis development and continue show a relatively normal retinal architecture after virus inoculation [41]. 

Additional studies by us, however, discovered that the MCMV-infected eyes of mice who are at an intermediate stage of MAIDS development, i.e., 4 weeks after retrovirus infection (MAIDS-4 mice), are resistant to retinitis development despite harboring high amounts of infectious virus (~30,000 infectious virus particles per eye) and at amounts equivalent to that found within the MCMV-infected eyes of retinitis-susceptible MAIDS-10 mice [41,43]. This unexpected finding prompted us to conclude that while virus replication and cytopathology indeed play a role in the pathogenesis of MAIDS-related MCMV retinitis, other virologic, immunologic, and/or pathogenic events must also contribute to the remarkable retinal tissue destruction observed in AIDS patients with HCMV retinitis. 

### 3.2. Humoral Immunity

Because seropositivity to HCMV is ~80% worldwide [4,6] and immunocompetent persons rarely develop HCMV retinitis [12,13], we initially hypothesized that humoral immunity must play a significant role in protecting healthy mice from developing retinal disease following subretinal MCMV inoculation. Accordingly, passive transfer studies were performed in which either hyperimmune anti-MCMV rabbit serum or neutralizing mouse anti-MCMV monoclonal antibodies were administered to retinitis-susceptible MAIDS-10 mice. To our surprise, passively administered anti-MCMV antibody neither reduced the frequency or severity of MCMV retinitis nor reduced the intraocular amounts of infectious virus [44,45]. Thus, antibody alone does not prevent or alter the pathogenesis of MCMV retinitis in mice with MAIDS, even when administered prior to subretinal MCMV inoculation. This unexpected finding using the MAIDS model of MCMV retinitis was later confirmed when a clinical trial using passively administered humanized monoclonal anti-HCMV antibodies to treat HCMV retinitis in patients with AIDS revealed no therapeutic effect [44,45], an observation that underscores the clinical relevance of the mouse model of MAIDS-related MCMV retinitis.

### 3.3. Cellular Immunity

Significant loss of cellular immunity and the concomitant appearance of opportunistic pathogens is a hallmark of AIDS development [46]. The appearance of HCMV retinitis during AIDS is no exception with retinal disease and subsequent vision loss commencing when CD4+ T-cell numbers become substantially low, i.e., below 50 CD4+ T cells per microliter of peripheral blood [40,47]. We therefore performed a series of studies using the MAIDS model of MCMV retinitis to define with some precision the relative role(s) of cellular immunity in protecting immunologically normal persons from HCMV retinitis development. The premise for these investigations was also supported by the observation that BALB/c mice treated with monoclonal antibodies to CD4+ or CD8+ T-cell subsets show increased susceptibility to retinitis following supraciliary MCMV inoculation [48,49]. 

Cytotoxic CD8+ T cells as a cellular component of adaptive immunity kills virus-infected cells via distinct pathways of cytolysis that include the perforin cytotoxic pathway and the Fas/Fas cytotoxic ligand pathway [50,51]. Because work by others showed that a reduction in total CD8+ T-cell numbers in BALB/c mice significantly increased susceptibility to MCMV retinitis [52,53], we initiated a more in depth investigation of the role of cytotoxic CD8+ T cells in protection against MCMV retinitis during MAIDS by focusing on the relative roles of the perforin cytotoxic pathway versus the FasL cytotoxic pathway in providing protection. This was accomplished using mice with a genetic deficiency in either the perforin cytotoxic pathway (PKO mice) or in the Fas/FasL cytotoxic pathway (*gld* mice). Upon subretinal MCMV inoculation of groups of PKO mice, *gld* mice, MAIDS mice, or immunocompetent mice, a clear difference in resistance or susceptibility to MCMV retinitis development was observed among the animal groups. Whereas mice without MAIDS but deficient in the Fas/FasL cytotoxic pathway showed absolute (0%) resistance to MCMV retinitis as observed for healthy mice who were immunocompetent, mice without MAIDS but deficient in the perforin cytotoxic pathway showed significant susceptibility to MCMV retinitis at a frequency of >90% and at a level equivalent to that observed for mice with MAIDS [50,51]. Adoptive transfer studies were then performed to confirm our observation that the perforin cytotoxic pathway is more important than the Fas/FasL cytotoxic pathway in conferring protection against MCMV retinitis [54,55]. Splenic immune cells originating from MCMV-immunized healthy mice or MCMV-immunized *gld* mice when transferred to recipient PKO mice significantly reduced susceptibility to MCMV retinitis following challenge by subretinal MCMV inoculation. In sharp contrast, splenic immune cells from donor MCMV-immunized PKO mice when transferred into recipient PKO mice failed to protect against MCMV retinitis upon challenge. 

The finding that loss of the perforin cytotoxic pathway, but not loss of the Fas/FasL cytotoxic pathway, imparts susceptibility to MCMV retinitis in mice without MAIDS prompted us to hypothesize that mice with MAIDS are susceptible to MCMV retinitis due to loss of the perforin cytotoxic pathway. This hypothesis was tested by comparing perforin mRNA levels of splenic T cells and MCMV-infected eyes collected from mice with MAIDS with perforin mRNA levels of splenic T cells and MCMV-infected eyes collected from healthy mice. In support of this hypothesis, perforin mRNA levels were dramatically reduced within splenic T cells and MCMV-infected eyes of retinitis-susceptible MAIDS mice when compared with retinitis-resistant healthy mice who showed consistently high amounts of perforin mRNA production within splenic T cells and MCMV-infected eyes [51,55]. 

That suppression of the perforin cytotoxic pathway during retrovirus-induced immunosuppression might represent a major mechanism by which mice with MAIDS become susceptible to onset of MCMV retinitis was explored further using a cytokine immunotherapy approach. Reasoning that both HIV-infected humans and LP-BM5-infected mice exhibit a profound systemic shift in cytokine production from a Th1 profile to a Th2 profile during progression of AIDS and MAIDS, respectively [56,57], we performed experiments whereby two Th1 cytokines, interleukin-2 (IL-2) and IL-12, were evaluated for their ability to restore protection against development of MCMV retinitis upon systemic administration. When administered a single intramuscular dose of polyethylene glycol (PEG)-administered human recombinant IL-2 prior to subretinal MCMV challenge, mice with MAIDS demonstrated an extraordinary reduction in the amount of infectious MCMV intraocularly as well as in the frequency of MCMV retinitis when compared with control mice with MAIDS not treated with PEG-IL-2 [58]. Indeed, PEG-IL-2 treatment provided absolute protection against the development of MCMV retinitis as underscored by a reduction in the frequency of retinal disease in untreated mice with MAIDS (~80%) to none (0%) in PEG-IL-2-treated mice with MAIDS. It is noteworthy that immunocompetent mice also show the same absolute protection against MCMV retinitis as seen in PEG-IL-2-treated MAIDS mice [59]. Surprisingly, however, intramuscular administration of PEG-IL-12 to mice with MAIDS prior to subretinal MCMV inoculation failed to reduce intraocular levels of infectious MCMV or frequency of MCMV retinitis when compared with control mice with MAIDS not treated with PEG-IL-12. Thus, two different Th1 cytokines that both stimulate cellular immunity [58] provided remarkably different outcomes when compared for their abilities to provide protection against MCMV retinitis during MAIDS. This study provides important proof-of-principle that cytokine immunotherapy will indeed significantly reduce the development of cytomegalovirus retinitis in the unique setting of retrovirus immunosuppression even though protection is cytokine specific. 

Having shown that loss of the perforin cytotoxic pathway results in susceptibility to MCMV retinitis [51] and that mice with MAIDS show greatly reduced levels of perforin mRNA within splenic T cells and MCMV-infected eyes [51], a logical extension of these findings was to determine if IL-2 immunotherapy restores resistance to MCMV retinitis during MAIDS by stimulation of the perforin cytotoxic pathway. This proved to be the case. Quantification of perforin mRNA levels within MCMV-infected eyes of mice with MAIDS administered PEG-IL-2 intramuscularly revealed a significant intraocular upregulation of perforin transcripts that correlated with a significant reduction in the frequency of MCMV retinitis from ~80% in untreated mice with MAIDS to 0% in PEG-IL-2-treated mice with MAIDS [58]. Moreover, IL-2 immunotherapy correlated with increased infiltration of CD8+ T cells within MCMV-infected eyes [60]. 

We therefore conclude that cellular immunity is far more important that humoral immunity in protecting immunocompetent mice from developing MCMV retinitis. Not only do cytotoxic CD8+ T cells play a prominent role in this protection, but it is the perforin cytotoxic pathway and not the Fas/FasL cytotoxic pathway of cellular immunity that is needed for protection. The perforin cytotoxic pathway appears to be lost in retinitis-susceptible mice with MAIDS but is restored by systemic IL-2 immunotherapy that results in renewed resistance to MCMV retinitis as seen in healthy mice. Whether dampening of the perforin cytotoxic pathway utilized by cellular immunity contributes to increased susceptibility to HCMV retinitis during AIDS remains to be determined. If true, measurement of perforin-mediated cytotoxicity might be a better quantification of cellular immunity than measurement of total CD4+ and/or CD8+ T cells as a predictor for risk of development of HCMV retinitis in various immunosuppressed patient populations including patients with AIDS [61]. 

### 3.4. Cytokines and Suppressor of Cytokine Signaling 1 and 3 

Our studies exploring cytokine immunotherapy with IL-2 or IL-12 encouraged additional investigations to learn more about the possible contributions of individual cytokines toward the pathophysiology of MAIDS-related MCMV retinitis. Initial studies focused on tumor necrosis factor alpha (TNF-a) and interferon gamma (IFN-g) because TNF-a is a major proinflammatory cytokine often associated with tissue necrosis [62] and IFN-g is a Th1 cytokine involved in the coordination of many immune responses [63]. To better understand their expression during the temporal progression of MCMV retinal disease, studies were performed using mice at different stages of retrovirus-induced immunosuppression and MAIDS development. Intraocular levels of TNF-a and IFN-g were therefore determined within the MCMV-infected eyes of mice at 2 weeks (MAIDS-2 mice), 4 weeks (MAIDS-4 mice), and 12 weeks (MAIDS-12 mice) after infection with the immunosuppressive LP-BM5 murine retrovirus mixture and compared with MCMV-infected eyes of age-matched immunocompetent mice. As predicted due to its association with tissue necrosis [41], intraocular levels of TNF-a progressively increased during the evolution of MAIDS and reached highest levels within the MCMV-infected eyes of MAIDS-12 mice when 100% of the animals exhibited retinitis [41]. In sharp contrast, intraocular levels of IFN-g progressively decreased as animals exhibited worsening immunosuppression and transitioned from healthy to mid-stage MAIDS to late-stage MAIDS when showing susceptibility to MCMV retinitis development [41]. Thus, we observed an inverse relationship between intraocular levels of TNF-a and IFN-g within MCMV-infected eyes during the evolution of retrovirus-induced immunosuppression as eyes became more susceptible to retinal necrosis development and the production of key Th1 cytokines simultaneously dampened. 

Additional studies were directed at IL-4 and IL-10, two major Th2 cytokines shown to increase systemically during the progression of HIV infection to AIDS as part of the Th1/Th2 cytokine shift [64]. Having shown that systemic cytokine immunotherapy with IL-2, a Th1 cytokine, restores protection against MCMV retinitis development in mice with MAIDS [65], we explored the possibility that systemic loss of IL-4 and/or IL-10 would conversely decrease the severity and/or frequency of MCMV retinitis. This proved not to be the case. Subretinal MCMV inoculation of mice with MAIDS who were also genetically deficient in either IL-4 (IL-4 ^−/−^ mice) or IL-10 (IL-10 ^−/−^ mice) production showed equivalently high intraocular amounts of infectious virus as well as equivalent susceptibility to MCMV retinitis when compared with wildtype mice with MAIDS [66]. These findings suggest that neither IL-4 nor IL-10 alone contribute to the pathogenesis of MAIDS-related MCMV retinitis, although the possibility that they function through other retrovirus-induced immunosuppressive pathways cannot be ruled out. 

IL-17 is another proinflammatory cytokine. Produced by CD4+ Th17 cells, this cytokine has been associated with several diseases of the eye including those of autoimmune origin [67]. Initial studies to determine the systemic fate of IL-17 production during the progression of MAIDS without MCMV infection revealed a progressive increase in IL-17 mRNA and protein production when comparing whole spleens recovered from healthy mice and from mice at 4, 8, and 10 weeks after immunosuppressive LP-BM5 murine retrovirus infection [68]. Systemic MCMV infection of these animal groups, however, resulted in a significant decrease in IL-17 production. This unexpected finding also extended to the pathogenesis of MAIDS-related MCMV retinitis; the intraocular levels of IL-17 mRNA and protein decreased significantly during MAIDS progression. Collectively, these findings support the fundamental idea that the productive replication of MCMV downregulates IL-17 production during MAIDS. The mechanism by which this occurs may involve IL-10 because IL-10 ^−/−^ mice with late-stage MAIDS exhibit partial restoration of IL-17 production in spleens or eyes following systemic or subretinal MCMV infection, respectively [68]. 

Alarmins are immune activating cytokines that are released from damaged cells. One such alarmin, IL-1a, functions early in the pathogenic process to induce inflammation via recruitment of macrophages and neutrophils. We therefore predicted that the MCMV-infected eyes of mice with MAIDS would show significant intraocular stimulation of IL-1a because macrophages and neutrophils are a prominent feature of the inflammation that develops during the pathogenesis of MAIDS-related MCMV retinitis (and AIDS-related HCMV retinitis). As predicted, retinitis-susceptible MCMV-infected eyes of MAIDS mice at 3 days after intraocular infection showed dramatic increases in IL-1a mRNA and protein amounts when compared with retinitis-resistant MCMV-infected eyes of healthy mice that showed minimal IL-1a production [69]. By comparison, retinitis-susceptible MCMV-infected eyes of mice immunosuppressed by corticosteroid treatment also showed significant intraocular increase in IL-1a production, albeit to a lesser extent than that observed for MCMV-infected eyes of MAIDS mice. Our findings, therefore, suggest a role for the alarmin IL-1a in the pathogenesis of MCMV retinitis in animals immunosuppressed by two distinct mechanisms, drug-induced immunosuppression and retrovirus-induced immunosuppression. 

Suppressor of cytokine signaling (SOCS) proteins are inducible negative regulators of cytokine signaling pathways [70,71]. SOCS1 and SOCS3 are the best studied of this family of host SOCS proteins. SOCS1 functions through the regulation of JAK/STAT proteins involved in IFN signaling [72], whereas SOCS3 is a negative feedback regulator of the IL-6 cytokine family [73]. The investigations described above that explored a number of individual cytokines vis-a-vis the pathogenesis of MAIDS-related MCMV retinitis prompted us to perform a series of in-depth studies to determine the extent that SOCS1 and/or SOCS3 might participate in the evolution of MCMV retinitis during MAIDS. The reader is directed to a comprehensive review of SOCS and herpesviruses with emphasis on cytomegalovirus retinitis for a more detailed summary of this work [74].

Initial studies revealed that SOCS1 and SOCS3 (but not SOCS5) mRNA and protein were remarkably stimulated within the MCMV-infected eyes of MAIDS-10 mice during retinitis development [75]. In comparison with the MCMV-infected eyes of animals immunosuppressed by LP-BM5 murine retrovirus infection, SOCS1 and SOCS3 were only moderately stimulated during less severe retinitis development within the MCMV-infected eyes of mice immunosuppressed by corticosteroid treatment [76]. Sources for SOCS1 and SOCS3 production within the retinas of MCMV-infected eyes of MAIDS-10 mice showing severe retinal necrosis proved to be resident microglial cells and Muller cells as well as infiltrating macrophages and granulocytes (neutrophils) [75]. Importantly, both virus infected and uninfected cells were found to produce SOCS1 and SOCS3. That uninfected cells were a source for these SOCS proteins is consistent with a bystander mechanism due to the simultaneous intraocular stimulation of the SOCS-inducing cytokines [61]. 

Additional studies were performed using an in vitro cell culture approach to define with greater clarity the production of SOCS1 and SOCS3 during the course of MCMV replication. Because macrophages contribute prominently to the inflammation observed during cytomegalovirus retinitis [77] and macrophages produce SOCS1 and SOCS3 during MAIDS-related MCMV retinitis [75], monolayers of mouse macrophages were used in these studies. As expected, MCMV infection of these cells resulted in the significant production of high amounts of SOCS1 and SOCS3 mRNAs. A biphasic pattern of SOCS mRNA production was observed with early SOCS1 and SOCS3 mRNA upregulation occurring before immediate early virus gene expression followed later by SOCS1 and SOCS3 mRNA upregulation at 3 days post-infection [78]. Inoculation of mouse macrophage cultures with replication-deficient UV-inactivated MCMV, however, failed to stimulate detectable SOCS1 or SOCS3 mRNA synthesis at all times examined suggesting that virus tegument proteins are insufficient for SOCS1 or SOCS3 stimulation within mouse macrophages. In sharp contrast, inoculation of monolayers of mouse fibroblasts with either infectious MCMV or UV-inactivated MCMV both resulted in the stimulation of SOCS1 and SOCS3 mRNA synthesis, but only early after inoculation [78]. Thus, the pattern of SOCS1 and/or SOCS3 mRNA stimulation during MCMV infection in cultured cells appears to be dependent upon cell type as well as times of synthesis relative to productive replication [78]. 

Collectively, the accumulated findings for SOCS1 and/or SOCS3 so far suggest that they may affect the severity of cytomegalovirus disease differentially in a cell-type-specific manner within the ocular compartment. It remains unclear, however, whether virus-induced stimulation of SOCS1 and/or SOCS3 is protective or pathogenic in the eye during MAIDS-related MCMV retinitis, an uncertainty that continues to motivate further investigations of SOCS and cytomegalovirus retinitis pathogenesis. Indeed, these host proteins are being recognized as possible therapeutic targets for cancers and other eye diseases and may therefore prove to be important for the future treatment of AIDS-related HCMV retinitis or other herpesvirus diseases [74]. 

### 3.5. Cell Death Pathways

Clearly, virus replication within retinal cells and the associated cytopathology contribute directly to retinal cell death during the onset and development of the necrosis that helps to define the unique retinal tissue pathology associated with AIDS-related HCMV retinitis. Not every retinal cell exhibits cytopathological changes, however, suggesting that other mechanisms of cell death may operate. One distinct possibility is the operation of programmed cell death pathways that serve to limit virus spread and confine pathology as part of the host’s innate immune responses against virus infection. Multiple cell death pathways have been recognized. We are therefore performing a series of investigations to determine the relative contributions, if any, of several individual cell death pathways toward retinal tissue destruction during the evolution of MAIDS-related MCMV retinitis. Those cell death pathways investigated to date include apoptosis, pyroptosis, necroptosis, and parthanatos (Table 1).

Recognized in 1972, apoptosis is probably the best studied of the cell death pathways, especially with respect to virus infections [79]. The mechanism by which apoptosis causes cell death is caspase-dependent (Table 1) and involves an extrinsic apoptotic pathway stimulated by TNF-a signal transduction (reviewed in [80,81]). Our prior discovery that intraocular levels of TNF-a increase significantly within MCMV-infected eyes during the progression of MAIDS [41] therefore lead us to first investigate apoptosis. In agreement with our previous finding [41], we detected not only significant amounts of TNF-a mRNA and protein within the MCMV-infected eyes of retinitis-susceptible mice with MAIDS, but also detected significant amounts of apoptosis-associated TNF receptor 1 (TNFR1), active caspase 3, and active caspase 8 [43]. Those cells found to produce TNF-a within retinal tissues of eyes at different times during the progression of MAIDS-related MCMV retinitis included microglial cells and Muller cells of the retina as well as macrophages and granulocytes (neutrophils) as inflammatory infiltrates. Mice with MAIDS deficient in TNF-a or TNFR1, and therefore deficient in the TNF-a extrinsic apoptotic pathway, were then used to determine the extent that apoptosis might contribute to the retinal tissue pathology of MCMV retinitis during retrovirus-induced immunodeficiency. Although the MCMV-infected eyes of mice with MAIDS deficient in TNF-a or TNFR1 showed decreased frequencies of retinitis when compared with the MCMV-infected eyes of wildtype mice, intraocular amounts of infectious virus remained the same for these MAIDS mice also deficient in the extrinsic apoptotic pathway when compared with wildtype MAIDS mice. More importantly, the histopathologic pattern of full-thickness retinal necrosis that developed in these animals was identical to that which developed within the eyes of wildtype mice with MAIDS [43]. Additional studies designed to quantify by TUNEL assay the precise contribution of apoptosis to retinal necrosis development revealed that only ~8% of retinal cells exhibited apoptotic activity within MCMV-infected eyes of mice with MAIDS at 6 days after intraocular MCMV inoculation [43]. Taken together, our findings suggest that apoptosis contributes minimally to the onset and development of MCMV retinitis during the pathogenesis of MAIDS. 

Pyroptosis is a more recently recognized caspase-dependent cell death pathway (Table 1) that is uniquely associated with the stimulation of a significant and sustained inflammatory response [82]. This is achieved through the cellular release of proinflammatory cytokines IL-1b and IL-18 via the formation of gasdermin D (GSDMD)-mediated membrane pores. To determine the possible pathogenic contribution of pyroptosis toward the development of full-thickness retinal necrosis during MAIDS-related MCMV retinitis, we initially demonstrated significant transcription and translation of all key pyroptosis-associated genes within the ocular compartments of MCVM-infected eyes of mice with MAIDS. These included caspase-1, GSDMD, IL-1b, and IL-18 [43] (Table 1). Subsequent investigations compared the MCMV-infected eyes of groups of wildtype mice with MAIDS with the MCMV-infected eyes of groups of mice with MAIDS deficient in either caspase-1, GSDMD, or IL-18. Histopathologic analysis revealed an expected full-thickness retinal necrosis in 100% of MCMV-infected eyes of wildtype MAIDS mice. In sharp contrast, none (0%) of MCMV-infected eyes of caspase-1 ^−/−^ MAIDS mice, GSDMD ^−/−^ MAIDS mice, or IL-18 ^−/−^ MAIDS mice developed full-thickness retinal necrosis. Instead, these animals exhibited an atypical pattern of retinal disease characterized by thickening and proliferation of the RPE layer with relative sparing of the neurosensory retina [83]. Surprisingly, MCMV-infected eyes of all groups of proptosis-deficient MAIDS mice harbored equivalent intraocular amounts of infectious virus as seen within MCMV-infected eyes of wildtype mice despite failure to develop full-thickness retinal necrosis. We therefore conclude that pyroptosis, unlike apoptosis, plays a significant role in the development of full-thickness retinal necrosis during the pathogenesis of MAIDS-related MCMV retinitis. 

More recent investigations have focused on two additional programmed cell death pathways, necroptosis [43] and parthanatos [84], that are both caspase independent pathways (Table 1). Evidence for the participation of necroptosis during the onset and development of MCMV retinitis during MAIDS has been provided by the detection of significant amounts of necroptosis-associated receptor-interacting protein kinase 1 (RIPK1) and RIPK3 (Table 1) mRNAs within the MCMV-infected eyes of retinitis-susceptible mice with MAIDS at different times during retinitis development [43]. Similarly, transcripts and protein products for key molecules that operate during the parthanatos pathway, poly (ADP-ribose) polymerase-1 (PARP-1), ADP-ribose (PAR), and poly (ADP-ribose) glycohydrolase (PARG)] (Table 1), have also all been found to increase significantly within the eyes of retinitis-susceptible mice with MAIDS following subretinal MCMV inoculation [84]. The precise contributions of necroptosis and parthanatos toward the development of MCMV-induced full-thickness retinal necrosis must await future investigations using mice with MAIDS that are deficient in one or more key pathway-associated genes as has been done for apoptosis and pyroptosis. 

### 3.6. Transcriptional Analysis of Immune Response Genes

Results of the many investigations summarized above underscore the unexpected complexity of the immunopathology of MAIDS-related MCMV retinitis that in many respects mirrors the complexity of retrovirus-induced immunopathology during the evolution of MAIDS in mice that also extends to AIDS in humans. With this in mind, we embarked on a new direction of investigation designed to assess in a more comprehensive manner the intraocular pattern of immune response gene expression within MCMV-infected eyes during health and disease. This was accomplished using NanoString nCounter technology that allows for the direct measurement of the transcription of hundreds of genes simultaneously and without the need for amplification [85]. We therefore used this technology to compare the intraocular expression of 575 immune response genes within the MCMV-infected eyes of immunologically normal mice and compared them with the MCMV-infected eyes of mice at different stages of MAIDS. As expected, our findings showed that intraocular MCMV infection of mice with MAIDS resulted in the robust upregulation or downregulation of a number of immune response genes that were associated with 32 distinct immunologic pathways during retinitis development at 3, 6, and 10 days post-infection. Moreover, as expected, the patterns of immune response gene activation were found to differ remarkably within MCMV-infected eyes of healthy mice resistant to retinitis development when compared with MCMV-infected eyes of mice at different stages of MAIDS that exhibit pronounced differences in their susceptibility to retinitis development [86]. Excluding genes that exhibited a consistent fold change of less than two, 17 genes were upregulated and 15 genes were downregulated within retinitis-resistant MCMV-infected eyes of healthy mice, 83 genes were upregulated and 4 genes were downregulated within MCMV-infected eyes of MAIDS-4 mice during the progression toward susceptibility to retinitis, and 92 genes were upregulated and 54 genes were downregulated within retinitis-susceptible MCMV-infected eyes of MAIDS-10 mice. A separate transcriptional analysis focused exclusively on programmed cell death pathways provided additional evidence for the involvement of pyroptosis and necroptosis during the pathogenesis of MAIDS-related MCMV retinitis (see Section 3.5 above). Thus, use of the NanoString nCounter analysis approach has therefore not only confirmed our previous findings, but has also remarkedly extended our understanding of the immunologic events that take place during the onset and development of full-thickness retinal necrosis within the MCMV-infected eyes of mice with retrovirus-induced immunosuppression. Collectively, our findings show that increased susceptibility to MCMV retinitis during the progression of MAIDS from healthy mice to MAIDS-4 mice to MAIDS-10 mice is associated with an unexpected robust upregulation or downregulation of a large number of immune response genes that are unique to each animal group investigated. Those immune response genes involved in this dynamic process include those associated with innate immunity, adaptive immunity, cytokine signaling, lymphocyte activation, and programmed cell death pathways. A complete summary of those individual differentially expressed immune response genes showing the greatest upregulation of activity within MCMV-infected eyes of healthy mice, MAIDS-4 mice, and MAIDS-10 mice is provided in our previous publication [85].

## 4. Conclusions

A number of clinically relevant mouse models of MCMV retinitis have been developed over the years that differ with respect to the method by which immunosuppression is achieved and with differing strengths and weaknesses [61]. Nonetheless, they collectively have provided new insights into the pathogenesis of AIDS-related MCMV retinitis and, toward this end, underscore the value of mouse models of MCMV retinitis. We have elected to focus on the MAIDS model of MCMV retinitis for this purpose as an experimental platform due to its retrovirus-induced immunopathology that mimics in many ways the immunopathology of HIV immunosuppression, especially the Th1/Th2 cytokine shift that is a prominent feature of MAIDS in mice and AIDS in humans. Table 2 summarizes our findings so far for those virologic, immunologic, and pathogenic mechanisms that appear to operate and perhaps essential for the development of full-thickness retinal necrosis in mice with MAIDS following intraocular MCMV infection. 

These findings bring particular attention to those events working either separately or in concert that could be exploited as new therapeutic approaches for the management of AIDS-related HCMV retinitis in the clinical setting. These findings might also extend to retinal diseases caused by other human herpesviruses in both immunologically normal persons and retrovirus-immunosuppressed patients. 

## Figures and Tables

**Table 1 pathogens-10-00850-t001:** Comparison of major programmed cell death pathways.

Cell Death Pathway	Caspase-Dependent?	Key Molecules
Apoptosis	Yes	TNF-a
		Caspases 3 and 8
Pyroptosis	Yes	Caspase-1
		Gasdermin D
		Inflammasomes
		Caspase-11
Necroptosis	No	RIPK1
		RIPK3
		MLKL
Parthanatos	No	PARP-1
		PAR
		PARG

**Table 2 pathogens-10-00850-t002:** Summary of virologic, immunologic or pathogenic events investigated or observed during MAIDS-related MCMV retinitis and relative contribution to retinal disease progression ^a,b^.

Virologic, Immunologic, or Pathogenic Event ^a^	Yes	No	Reference
Virus replication and cytopathology	XX		[29]
Humoral immunity	XX		[44]
Loss of perforin-mediated cytopathology	XX		[51,55]
Loss of Fas/Fas ligand-mediated cytopathology		XX	[51,55]
Events leading to reduction in retinal disease:			
-IL-2 immunotherapy	XX		[51,58,60]
-IL-12 immunotherapy		XX	[58]
-Loss of IL-4		XX	[38,66]
-Loss of IL-10		XX	[66,68]
Intraocular stimulation of cytokines or signaling pathway molecules:			
-TNF-α	XX		[41,43]
-IFN-γ		XX	[41]
-IL-4	XX		[65,66]
-IL-10	XX		[66,68]
-IL-17		XX	[68]
-IL-1α (alarmin)	XX		[69]
-SOCS1	XX		[68,75,76]
-SOCS3	XX		[68,75,76]
-SOCS5		XX	[78]
Intraocular stimulation of cell death pathway molecules:			
-Apoptosis	XX		[43]
-Necroptosis	XX		[43]
-Pyroptosis/Inflammasomes	XX		[14,43]
-Parthanatos	XX		[84]
Loss of cell death pathway results in atypical retinal disease:			
-Apoptosis		XX	[43]
-Pyroptosis	XX		[83]

^a^ Modified from [42]; ^b^ Excludes transcriptional analysis of immune response genes [86].

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
