# Peer review of "A Mouse Model That Mimics AIDS-Related Cytomegalovirus Retinitis: Insights into Pathogenesis"

_pathogens, 2021, doi:10.3390/pathogens10070850_

Round 1

Reviewer 1 Report

This review article is excellently organized and the content is very informative and the assessment accurate.

I would suggest stating in section 3.6. (the transcriptional analysis of immune response genes is discussed) what are the main findings? At the moment it is stated how the analysis was performed, and how many genes were dysregulated. However, there is no discussion on specific results that were obtained and the 'take-home message' of this analysis. 

Author Response

We thank this reviewer for favorable comments that included “this review article is excellently organized and the content is very informative and the assessment accurate”. Our response to the one helpful suggestion provided by this reviewer is provided below.

  1.   I would suggest stating in section 3.6 (the transcriptional analysis of immune response genes is discussed) what are the main findings? At the moment it is stated how the analysis was performed, and how many genes were dysregulated. However, there is no discussion on specific results that were obtained and the ‘take-home message’ of this analysis.

Response: Additional information (see below) is now provided in the revised manuscript regarding specific results obtained during our transcriptional analysis of immune response genes during the onset of progression of MAIDS-related MCMV retinitis.

“A separate transcriptional analysis focused exclusively on programmed cell death pathways provided additional evidence for the involvement of pyroptosis and necroptosis during the pathogenesis of MAIDS-related MCMV retinitis (see Section 3.5 above). Thus, use of the NanoString nCounter analysis approach has therefore not only confirmed our previous findings, but has also remarkedly extended our understanding of the immunologic events that take place during the onset and development of full-thickness retinal necrosis within the MCMV-infected eyes of mice with retrovirus-induced immunosuppression. Collectively, our findings show that increased susceptibility to MCMV retinitis during the progression of MAIDS from healthy mice to MAIDS-4 mice to MAIDS-10 mice is associated with an unexpected robust upregulation or downregulation of a large number of immune response genes that are unique to each animal group investigated. Those immune response genes involved in this dynamic process include those associated with innate immunity, adaptive immunity, cytokine signaling, lymphocyte activation, and programmed cell death pathways. A complete summary of those individual differentially expressed immune response genes showing the greatest upregulation of activity within MCMV-infected eyes of healthy mice, MAIDS-4 mice, and MAIDS-10 mice is provided in our previous publication [85].”

Reviewer 2 Report

In HIV patients due to immunosuppression of immune system by the virus, HCMV infection of AIDS patients may cause severe retinal disease that can leads to loss of vision. Early investigations of AIDS-related HCMV retinitis was difficult to pursue due to lack of a proper animal model. However, later on several mouse models of experimental cytomegalovirus retinitis were developed. These animal models provided a great insight into the pathophysiology of HCMV retinitis during AIDS. In this well written review, the authors have provided a detailed and in depth comparison of different mouse models of AIDS-related HCMV retinitis.

Overall, this is an excellent review of AIDS-related HCMV retinitis. The review is well written, and well structured. This review will be of great interest to general audience and to those who specifically work on AIDS-related HCMV retinitis. Although this review is very interesting, there are few minor points that need to be addressed before publishing this review as follow:

  1. Line 278: “that” should be “than”.
  2. Can the authors comment about the relevance of MCMV to HCMV?
  3. Is there any sex differences between male versus female mice with regards to AIDS-related HCMV retinitis?
  4. Is AIDS-related HCMV retinitis unilateral or bilateral?

Author Response

We thank this reviewer for the many favorable comments that included “this is an excellent review” that is “well written and well structured”. Specific responses to the four helpful suggestions provided by this reviewer are provided below.

  1. Line 278: “that’ should be “than”.

Response: Line 278 of the original manuscript states “MCMV challenge, mice with MAIDS demonstrated an extraordinary reduction in the” and does not contain the word “that”. We therefore cannot provide a revision to correct a typographical error within Line 278 as requested by Review #2. We are happy to comply with the requested revision if another sentence within the original manuscript is provided by this reviewer.

  1. Can the authors comment about the relevance of MCMV to HCMV?

Response: This requested information is already provided within the original manuscript. See Lines 88 – 92 with References 22 and 23 provided.  

  1. Is there any sex differences between male versus female mice with regards to AIDS-related HCMV retinitis?

Response: We have added “affecting both males and females without sex differences” to the revised manuscript at Line 74 of the original manuscript.

  1. Is AIDS-related HCMV retinitis unilateral or bilateral?

            Response: We have added “either unilaterally or bilaterally” to the revised manuscript at Line 70 of the original manuscript.